# OpenReview forum: "Heterogeneous Value Alignment Evaluation for Large Language Models"
_ICLR.cc/2024/Conference — ICLR 2024 Conference Withdrawn Submission_

### Official Review · Reviewer_urTP · 2023-10-29

**Soundness:** 2 fair
**Presentation:** 2 fair
**Contribution:** 2 fair
**Rating:** 3
**Confidence:** 5

**Summary:**

This paper uses an evaluation suite, the Social Value Orientation (SVO) slider measure, to assess the value alignment of LLMs. They instruct the LLMs to follow a certain value (individualistic, prosocial, altruistic, and competitive) and ask the LLMs to answer questions in the SVO slider measure. They assess how well the LLMs align to the instructed value. They call the evaluating framework "heterogeneous value alignment evaluation (HVAE)". They conducted experiments on five LLMs and showed that GPT-4 can align to the value the best, while most models tend to show prosocial regardless of the given value. They also discuss why the LLMs fail to align with the given value.

**Strengths:**

- Originality: Using SVO slider measure to evaluate the value of LLMs is an interesting topic.

**Weaknesses:**

1.  The experiment setting is not clearly specified.
    - How the LLMs are prompted is not clearly elaborated in the text. The paper only includes Figure 3 without explaining it in the main text. It is also unclear how the LLMs are instructed to follow a certain value.
    - How the $D$ shown in Equation (3) is designed is not clearly specified.
    - How the $v_{target}$ ins Equation (3) is constructed is not clearly specified.
    - The version of GPT-3.5/4 models used in the paper is not specified.

2. The experiment setting cannot really reflect if the model is able to align to a certain value.
    -  **The model can misunderstand the meaning of a value**: Currently, the authors give the LLM a certain value (individualistic, prosocial, altruistic, and competitive) and ask the LLMs to list the trait of the value. In Section 4.3, they show that the LLMs can generate traits that do not correspond to the value. This shows that the results in this paper are largely affected by whether LLMs understand the meaning of the value. I do not think this is really what we want to evaluate. A more reasonable setting should be using the precise definition of these values (perhaps looking up the dictionary) to instruct the LLMs about the value, instead of asking the LLMs to infer what traits should be included in those values. In fact, I also do not know what those values mean and I have to search the web to know what those values mean and what are the traits of a specific value.
   - **Testing set contamination**: It is highly possible that LLMs are trained on a dataset that contained SVO slider measurement. So when the LLM is asked to follow a certain value, it is possible that they simply memorize how a person with a certain value answers the question. This is like evaluating the IQ of someone who knows the ground truth answer of the IQ test. So, It is possible that the LLM only knows that mapping between a value and its corresponding answering pattern even if it does not know what the value really means and what traits the value should have. Even if the LLMs show alignment on SVO slider measurement does not imply that they are really aligned with the value. I prompt GPT-4 and ask if it knows the SVO slider measurement, and it can perfectly explain what that is, and it also knows how to answer to fulfill a specific value. I do not think this is what we want to measure.

3. Insufficient experiments. LLMs are notorious for their variance due to prompts. I would like to know if the results change if we put the value in the system prompt instead of putting it in the dialogue.

4. Some claims are not well-supported by the experiments.
     - Section 4.3: Scaling laws: The models used in this paper differ not only in the parameter sizes but also in the training data. This makes discussing the scaling law here unreasonable.
    - `Considering we don’t know the training details of the two closed-source models GPT-4 and GPT-3.5, one possible reason for the prosocial orientation is the use of Safe-RLHF technology, which can be found in Llama2`. This is odd since Claude, in Figure 8, shows a similar pattern to GPT-4 and Claude is also trained with safe-RLHF, or precisely, constitutional AI. So the hypothesis on Safe-RLHF is not supported by Figure 8.

**Questions:**

1. What does the goal prompt mean and what does "without goal prompt" in Figure 8 mean?

---

### Official Review · Reviewer_eMvm · 2023-10-30

**Soundness:** 2 fair
**Presentation:** 3 good
**Contribution:** 2 fair
**Rating:** 3
**Confidence:** 4

**Summary:**

This paper integrates the concept of Social Value Orientation (SVO) from social psychology to evaluate the alignment of Large Language Models (LLMs) across four dimensions: Altruistic, Individualistic, Prosocial, and Competitive. The methodology involves prompting LLMs to articulate the definition of each social perspective and subsequently instructing them to distribute a set of coins between two groups, one for themselves and the other for another party. The degree to which LLMs embody a particular perspective is ascertained by evaluating the consistency between their provided definitions and their decisions in the coin distribution task. The experimental analysis includes two proprietary and three open-source models using this assessment framework. The findings indicate that GPT-4 outperforms others, yet there is no definitive correlation between model scale and experimental performance.

**Strengths:**

- Sought to incorporate a social psychology theory into the assessment framework for Large Language Models (LLMs).
- Presented a fresh approach to evaluating LLMs.

**Weaknesses:**

- The proposed viewpoint is clear and intuitive; however, the scope of the associated research activity and area may be too limited for publication as a full paper in ICLR. It is advisable to expand the content related to the proposed framework, such as detailing ways to use the framework to enhance or more accurately evaluate specific applications.
- The connection between the theoretical discourse in Section 3 and the practical methodologies applied in the experimental procedures of Section 4 is somewhat ambiguous.

**Questions:**

- Figure 5 reveals that none of the models evaluated aligned with the perspectives at the radial extremes of the SVO sphere, namely the Altruistic and Competitive orientations. This could suggest that the models are inherently incapable of aligning with such polarized perspectives, or it may reflect a bias in the evaluation metric that favors moderate values over extreme ones. Further investigation into this phenomenon would be beneficial to ascertain the underlying causes.

---

### Official Review · Reviewer_P3tB · 2023-10-31

**Soundness:** 3 good
**Presentation:** 3 good
**Contribution:** 2 fair
**Rating:** 3
**Confidence:** 4

**Summary:**

The paper studies whether or to what extent LLMs can align with heterogeneous values. To that end, the authors propose the "value rationality" that measures the alignment degree between an agent’s behavior and a target value. Then, they develop a pipeline named HVAE (Heterogeneous Value Alignment Evaluation) that utilizes social value orientation (SVO) to quantify agents’ value rationality. Based on the proposed framework, they evaluate the value rationality of five mainstream LLMs.

**Strengths:**

1. The motivation is clear and interesting. It would be a promising direction to align LLMs with diverse and dynamic values. This paper is a good try on the direction.

2. With the increasing capability of LLMs, integrating social psychology would be a natural and interesting fashion and this work is also one of the pioneers in this direction.

**Weaknesses:**

1. The experiments are simple and provide few insights. At its core, the main experiment is to query several LLMs with the SVO options in different contexts. The results can also be interpreted as whether or to what extent LLMs can role-play the four roles in the SVO system.

2. As noticed in the paper, aligning specific values by in-context prompting LLMs (the method used in the paper) can be limited. Most tested LLMs (e.g., GPT-4, GPT-3.5, Llama2-Chat) have been already aligned with some static values, making the prompting methods used in this work more like a weak "post-alignment". Therefore, the obtained results may be limited.

3. The SVO test is built upon the capabilities of instruction-following and basic reasoning while some models (e.g., Llama2 and Vicuna) are not fully capable of such abilities. Therefore the SVO test is a little ahead of the current time, or one should include more capable models.

**Questions:**

N/A

---

### Official Review · Reviewer_UkcL · 2023-11-02

**Soundness:** 2 fair
**Presentation:** 2 fair
**Contribution:** 2 fair
**Rating:** 3
**Confidence:** 4

**Summary:**

Paper addresses the value alignment problem in LLMs and proposes to implement social value orientation evaluation from social psychology. Study presented in the paper is generally complete but the chosen objective and problem, the definitions related and the chosen methodolgy are not well supported and is difficult to align. A revision on clarifying the objectives and motivation will help better assess the paper and its results.

**Strengths:**

Introduces new approach to evaluate value alignment in LLMs

**Weaknesses:**

The object of the study and chosen methodology do not align.
The related work to the paper is not well connected.
Paper's area of study in psychology and how the chosen terminology is adapted is not well supported.

**Questions:**

Does heterogeneity really define the relativity in importance of weights to specific features in values?
How does subjectivity and the definition of heterogeneity described in the introduction overlap logically?